# Estimation of the Relative Chlorophyll Content of Pear Leaves Based on Field Spectrometry in Alaer, Xinjiang

**DOI:** 10.3390/s25113552

**Published:** 2025-06-05

**Authors:** Yufen Huang, Zhenqi Fan, Hongxin Wu, Ximeng Zhang, Yanlong Liu

**Affiliations:** 1College of Information Engineering, Tarim University, Alaer 843300, China; 19316400746@163.com (Y.H.); wuhongxin426@gmail.com (H.W.); zhangxmxixixi@sina.com (X.Z.); 2Key Laboratory of Tarim Oasis Agriculture, Ministry of Education, Tarim University, Alaer 843300, China; 3College of Cyber Security, Tarim University, Alaer 843300, China; 4College of Agriculture, Tarim University, Alaer 843300, China; 18699423383@163.com

**Keywords:** discrete wavelet, relative chlorophyll content, quantitative inversion, partial least squares

## Abstract

Leaf chlorophyll content is an important indicator of the health status of pear trees. This study used Korla fragrant pears, a Xinjiang regional product, to investigate methods for estimating the relative chlorophyll content of pear leaves. Samples were collected from pear trees in the east, south, west, and north positions of peripheral canopy leaves. The leaf soil plant analysis development (SPAD) method was implemented using a SPAD-502 laser chlorophyll meter. The instrument measures the relative chlorophyll content as the SPAD value. Leaf spectra were acquired using a portable field spectrometer, ASD FieldSpec4. ViewSpecPro 6.2 software was employed to smooth the ground spectral data. Traditional mathematical transformations and the discrete wavelet transform were used to process the spectral data, then correlation analysis was employed to extract the sensitive bands, and partial least squares regression (PLS) was used to establish a model for estimating the chlorophyll content of pear tree leaves. The findings indicate that (1) the models developed using the discrete wavelet transform had coefficients of determination (*R*^2^) exceeding 0.65, and their predictive performance surpassed that of other models employing various mathematical transformations, and (2) the model constructed using the L1 scale for the discrete wavelet transform had greater estimation accuracy and stability than models established through traditional mathematical transformations or the high-frequency scale for discrete wavelet transform, with an *R*^2^ value of 0.742 and a root mean square error (RMSE) of 0.936. The prediction model for relative chlorophyll content established in this study was more accurate for chlorophyll monitoring in pear trees, and thus, it provided a new method for rapid estimation. Moreover, the model provides an important theoretical basis for the efficient management of pear trees.

## 1. Introduction

[Research significance] Chlorophyll is employed by plants to acquire, transform, and fix solar energy, and the chlorophyll content of leaves is an important indicator of the growth and development status of crops [1,2,3]. Therefore, accurately determining the chlorophyll content of crops is of theoretical and practical significance. [Previous Research Progress] As an important analytical tool, spectroscopy can accurately estimate the chlorophyll content of plants [4,5,6]. Spectroscopy provides comprehensive information about sample properties through wavelength-specific absorption, reflection, and emission characteristics [7,8,9], while enabling simultaneous acquisition of multiple features for more holistic analysis [10,11]. Hyperspectral technology has emerged as a powerful tool for monitoring plant physiological parameters due to its efficiency and non-destructive nature, where optimal wavelength selection and spectral band combinations are crucial for enhancing model accuracy. Although spectral features (e.g., red-edge bands) enable chlorophyll estimation [12,13,14], the optimal wavebands for specific crops exhibit significant differences. For instance, the chlorophyll-sensitive wavebands in cotton are concentrated within the range of 670–770 nm [15,16], whereas for rice leaves, wavebands around 620 nm and 705 nm are more sensitive to chlorophyll content [17,18]. Precise band selection via hyperspectral analysis thus supports early detection of crop stresses and nutritional deficiencies. Yu and Feng et al. [19] established a hyperspectral inversion model for japonica rice nitrogen (N) content using ASD-acquired (Model A122325, ASD Inc., Malvern Panalytical, Longmont, CO, USA) reflectance at 500, 566, 600, and 712 nm wavelengths that showed strong correlations with N concentration. Singhal, G et al. [20] developed a chlorophyll concentration estimation model for maize leaves by identifying optimal spectral bands in the red region through the integration of multispectral UAV imagery and ground-based hyperspectral data. Tian et al. [21] demonstrated distinct sensitive spectral bands for leaf nitrogen accumulation (LNA) and concentration (LNC) in rice canopies using FieldSpec Pro FR 2500 spectroradiometer measurements, revealing the importance of parameter-specific band selection for N monitoring. While ongoing advancements in spectroscopic instrumentation continue to refine hardware capabilities, parallel developments in spectral processing methodologies are proving equally critical for achieving precise chlorophyll content estimation. Rajaee T et al. [22] combined the discrete wavelet transform (DWT) with an artificial neural network (ANN), multiple linear regression (MLR), and genetic algorithm-supported vector regression (GA-SVR) models for the prediction of chlorophyll-a levels. Zhang L et al. [23] targeted the absorption and reflection characteristics of chlorophyll concentration in inland waters, utilizing DWT to remove noise from water spectra. Subsequently, they extracted chlorophyll-sensitive bands through gray-level correlation analysis and employed machine learning to develop a predictive method for total phosphorus (TP) concentration in large lakes. Li F. et al. [24] took winter wheat as an example and conducted DWT analysis on the original canopy spectra, log-transformed spectra, first-derivative spectra, and continuum-removed spectra, respectively, to investigate the quantitative relationship between LNC and characteristic parameters. They also compared their results with models established using reflectance at sensitive wavelengths and typical spectral indices. The findings indicated that incorporating DWT analysis could effectively extract sensitive wavelengths from the spectra, thereby improving the prediction accuracy of the models. [Entry point of this study] Although there are many studies concerning the estimation of chlorophyll content, most of the current research is on general crops or those unique to specific provinces, and few studies have been conducted on cash crops unique to the Xinjiang region. Xinjiang has a long history of cultivation of Korla fragrant pears, one of the most important cash crops in the Xinjiang region, and estimation of leaf chlorophyll content is important for precise fertilizer application and field management [25,26]. [Key problems to be solved] Using field spectrometry technology, we applied a correlation algorithm to the spectral data of Korla fragrant pear leaves and leaf chlorophyll content, extracted the most sensitive bands, and constructed an estimation model for the chlorophyll content of Korla fragrant pear leaves.

## 2. Materials and Methods

### 2.1. Overview of the Pilot Study Area

The pear plantation based at Tarim University in Alaer City, Xinjiang Uygur Autonomous Region, was used as the test area. The plantation extends from the southern foothills of the Tianshan Mountains in the north to the northern edge of the Taklamakan Desert in the south [27]. The area has flat terrain with a height difference of 1/3000. The geospatial location of the plantation is shown in Figure 1. The reclamation area has an extreme continental arid desert climate, being located within the warm temperate zone, with low rainfall, little snow in winter, and strong surface evaporation, with an average annual precipitation of 40.1–82.5 mm and an average annual evaporation of 1876.6–2558.9 mm [28,29]. The average annual solar radiation is 133.7–146.3 kcal/cm^2^, and the average annual sunshine is 2556.3–2991.8 h, with the percentage of sunshine at 58.69%. The reclamation area belongs to the Tarim River Plain, and the soil has been developed into irrigation meadow soil after years of reclamation [30]. Due to geographical conditions and climatic factors, Alaer is prone to sand-dust and floating-dust weather during spring and summer [31,32,33,34].

### 2.2. Research Methodology

#### 2.2.1. Pear Tree Canopy Sample Collection

The experiment was conducted on the canopy leaves of pear trees during the fruiting stage. Twelve Korla fragrant pear trees with essentially the same crown width and branching, equal trunk and branch thickness, no pests or diseases, and normal fruiting status were selected as the experimental objects. This selection was made to ensure that SPAD values would not be influenced by these factors. The sampled leaves were carefully chosen from the expansion period and were located in the east, south, west, and north directions of the canopy. Specifically, the leaves were collected from the middle of the current year’s branches in the periphery of the canopy. In the final collected samples, there were 22 leaves from the east side, 22 from the south side, 22 from the west side, and 21 from the north side. Figure 2 presents the sample images.

#### 2.2.2. Spectroscopy and Determination of the Relative Chlorophyll Content

The front lens of a portable geophysical spectrometer, ASD FieldSpec4 (Malvern Panalytical, Longmont, CO, USA), was set as a diffuse reflective 25° field-of-view probe. The system configuration of the instrument was optimized, and the whiteboard was corrected before the determination of spectrograms. To ensure that the field-of-view range was centered on the leaf, a distance of 2–3 cm was maintained between the spectrometer and the sample leaf during the measurements. The sunny side of the experimental leaves was used as the measurement surface, and the arithmetic mean of 20 spectra was used as one sampling spectrum. During the testing process, the instrument was optimized and calibrated every 10 min. The chlorophyll content of the experimental leaves was determined immediately after spectral data acquisition using a SPAD-502 laser chlorophyll meter (Konica Minolta, Tokyo, Japan). The SPAD-502 emits light beams of specific wavelengths to penetrate leaf tissue, leveraging the selective absorption characteristics of chlorophyll at these wavelengths. It quantifies changes in transmitted light intensity and converts them into chlorophyll concentration values via an algorithm. Each measurement takes only 1–2 s, enabling rapid monitoring of relative chlorophyll content in samples with a measurement error of just ±1.0 SPAD units [35]. Prior to measurement, clean the leaf surface to ensure no dust or other substances interfere with the results. During measurement, first select five evenly distributed measurement points on the leaf, avoiding damaged, senescent, or abnormal areas. Gently clamp the instrument’s probe onto the selected points, avoiding leaf veins and ensuring full coverage of the probe by the leaf. Finally, wait for the reading to stabilize before recording the SPAD value, and calculate the average from the 5 measurement points per leaf. Table 1 presents the sample statistics.

#### 2.2.3. Processing of Spectral Data

Due to the unique climatic conditions in Alaer, the collected pear tree leaves tend to accumulate dust, which results in a significant amount of noise being present in the leaf spectral data. Median filtering using ViewSpecPro 6.2 software was used to smooth the spectral data. The inverse transform, logarithmic transform, inverse first-order differential transform, logarithmic first-order differential transform, and DWT were then used to transform the smoothed data. Traditional mathematical transform methods for processing spectral data can provide useful information and suppress noise in the spectrum. The wavelet transform, developed from the Fourier transform, is a processing algorithm that can be applied to the signal at multiple scales to extract information within the signal. Wavelet transforms can be divided into continuous and discrete forms, where the DWT can separate low-frequency and high-frequency information within the spectrum scale, thereby improving the signal-to-noise ratio of the original spectrum. MATLAB R2022a software is used to perform three-layer decomposition on the original frequency spectrum of DWT using the Haar wavelet basis, where L and H represent low-frequency decomposition and high-frequency decomposition, respectively. The result is shown in Figure 3.

#### 2.2.4. Construction of a Model for Estimating the SPAD Value of Pear Leaves and Validation of Model Accuracy

To accurately reflect the characteristics of the estimation methods for relative chlorophyll content, the leaf spectral data were processed using the traditional spectral transform and DWT, and the spectral data were correlated with the relative chlorophyll content using the IBM SPSS Statistics 26 software to screen and extract sensitive bands. The estimation model for the relative chlorophyll content of leaves was established using the PLS algorithm on the processed data using MATLAB R2022a. To evaluate the estimation model, the values of the coefficient of determination (*R*^2^) and the RMSE were chosen as the evaluation indexes. The respective calculation methods are given in Equations (1) and (2). To estimate the relative chlorophyll content, the samples were randomly divided into two groups: a modeling group (60 values) and a validation group (27 values). The modeling group was used to construct a model for the estimation of the relative chlorophyll content of pear leaves, whereas the sample data from the validation group were used to assess the performance of the model.(1)$$R^{2}=1-\frac{\sum_{i=1}^{n}{\left(CHL_{i}-CHLP_{i}\right)}^{2}}{\sum_{i=1}^{n}{\left(CHL_{i}-\bar{CHL}\right)}^{2}}$$(2)$$RMSE=\sqrt{\frac{\sum_{i=1}^{n}{\left(CHL_{i}-CHLP_{i}\right)}^{2}}{n}}$$
where CHL is the relative chlorophyll content of leaves; CHLP is the estimated value based on the estimate from the model; and $\bar{CHL}$ is the mean value of the measured relative chlorophyll content.

## 3. Results

### 3.1. Correlation Between the Traditional Mathematically Transformed Spectra and the SPAD Value of Pear Leaves

The distribution of the coefficient of determination (*R*^2^) between the raw spectral data (R) as well as after five conventional data transforms (the inverse transform (1/R), the logarithmic transform [LOG(R)], the first-order differential transform (R′), the inverse’s first-order differential transform [(1/R)′], and the logarithmic first-order differential transform [LOG(R)′]) and the relative chlorophyll content of the leaf is illustrated in Figure 4 (X: λ (nm); Y: Transform Method; Z: *R*^2^).

The sensitive regions of the inverse transform, logarithmic transform, and first-order differential of the inverse were consistent with those of the original spectra, being primarily located in the 400–700 nm bands, but were generally coarser. These two transformations did not significantly improve the sensitivity. In contrast, the sensitive bands in the first-order differential and logarithmic first-order differential transforms were more dispersed, with almost no concentrated areas. These sensitive regions are distributed across the visible light, near-infrared, and partial infrared spectral bands. In the near-infrared region, the sensitivity bands of first-order differentiation transformations and first-order logarithmic differentiation transformations were similar. In summary, the correlation between the spectra of the first-order differential transform and the relative chlorophyll content of pear leaves was the lowest, whereas the sensitive region of the logarithmic transform was relatively more concentrated, and the distribution was more concentrated in the inverse transform and the inverse first-order differential transform.

### 3.2. Correlation of Discrete Wavelet Transformed Spectra with SPAD Value of Pear Leaves

After DWT (Figure 5 (X: λ (nm); Y: DWT Transform; Z: *R*^2^)), the roughness of the decision coefficient matrix plot increased gradually with the decomposition scale, and the resolution of the spectrum decreased by a factor of two for each increase in the scale of the DWT. The bands sensitive to relative chlorophyll content were largely distributed among the blue, green, and red light wavelengths in the visible region scale L1. This sensitive region did not change significantly with an increase in the decomposition scale, but the texture of the sensitive region in the blue-light region tended to be smoothed. This was because the spectral resolution gradually decreases after discrete wavelet processing, helping to suppress the effect of noise and improving the signal-to-noise ratio of low-frequency information. In contrast, the distribution of the sensitive bands was relatively dispersed in the H1–H3 scale, covering the visible and near-infrared regions, and there was a small shift with the increase in the decomposition scale. In summary, the correlation between the spectra and the relative chlorophyll content of pear leaves was high after the DWT treatment. The DWT reduced the influence of noise on low-frequency information, thereby improving the signal-to-noise ratio of the spectral information.

### 3.3. A Model for Estimating the SPAD Value of Pear Leaves

#### 3.3.1. Constructing a Model for Estimating the SPAD Value of Pear Leaves Based on Mathematical Transformations

The models constructed based on the inverse transform, the logarithmic transform, the first-order differential transform, the first-order differential transform of the inverse, and the first-order differential transform of the logarithm are listed in Table 2. The logarithmic model was the optimal model, with an estimation accuracy of *R*^2^ = 0.696 and an RMSE of 1.095. The characteristic waveband of the selected model was located in the visible band, and chlorophyll had a strong reflectance peak in the visible band. The spectral reflectance was primarily due to multiple reflections from chlorophyll, anthocyanins, and carotenoids within the leaves [4]. In summary, the logarithmic transformation can be used to estimate the relative chlorophyll content of the leaves of Korla fragrant pear trees.

#### 3.3.2. Construction of a Diagnostic Model for Estimating the SPAD Value of Pear Leaves Based on the Discrete Wavelet Transform

A comparative analysis of the DWT (Table 3) indicated that the model constructed based on low-frequency information was more stable than the model constructed based on high-frequency information. With an increase in the decomposition scale (L1–L3), the accuracy of the model gradually decreased, and the estimation accuracy of the relative chlorophyll content of pear leaves showed a trend of initially increasing and then decreasing. The main reason for this was that as the scale increases, the information in the original spectrum is gradually lost and is retained in the low-frequency bands, resulting in a gradual decrease in the available information. The model constructed based on L1 showed relatively high accuracy and stability and thus was considered the optimal model, with an estimation accuracy of *R*^2^ = 0.742 and an RMSE of 0.936. The characteristic bands were located in the blue and red regions of the visible region, i.e., the strong absorption regions of chlorophyll. For the H1–H3 model, the estimation accuracy showed a decreasing and then an increasing trend with an increase in the decomposition scale. However, the stability of the model constructed based on H1–H3 was poor, as it was affected by the moisture content of the study area. This resulted in a gradual decomposition of the noise information caused by air impurities into individual high-frequency information, which seriously affected the stability of the model. In summary, due to the influence of atmospheric noise, models constructed based on high-frequency information had poor stability, while models constructed based on low-frequency information were more stable, with the model constructed based on L1 performing the best.

### 3.4. Comparative Analysis of Relative Chlorophyll Content Regression Models for Pear Leaves

To compare the different modeling methods, the best-performing model among the different methods was selected, and a regression model was constructed using Origin 2021. The regression model with the DWT showed the highest value of *R*^2^ and the smallest RMSE for the validation set (Table 3). The corresponding model had the highest prediction accuracy. For the logR transform, the model with a single independent variable performed better than the multivariate model. This was because the characteristic wavelet coefficients extracted from the DWT model were more concentrated at the wavelength position, and the spectral signal could be decomposed into frequency components scale by scale, thus effectively separating the noise from the target information in the original spectral data. However, there was a greater fluctuation of the logR curve, causing the multivariate model to be more easily affected by the background conditions and thus reducing the performance. Figure 6 presents the linear regression results for the relative chlorophyll content inversion model. In summary, the regression model constructed by the discrete wavelet analysis method had excellent performance in estimating relative chlorophyll content, and the logR model also performed well among the traditional mathematical models.

## 4. Discussion

In this study, Korla fragrant pears grown in Alaer City were selected as the object of study, and the characteristics of relative chlorophyll content in the leaves of the pear trees were investigated and analyzed. During the growth of pear trees, plant biochemical parameters related to the physiological state and growth stage of the tree result in differences in the spectral responses. Therefore, spectral data can be transformed to highlight or enhance specific features or information in the data [36,37]. In this study, the high-frequency information from the discrete wavelet performed poorly in enhancing the sensitivity of the spectrum to the relative chlorophyll content of Korla fragrant pear leaves. The *R*^2^ and RMSE values of models established through high-frequency information were inferior to those established through low-frequency information. This was likely due to the prevalent dusty and sandstorm-prone weather conditions characteristic of the Alaer region, as well as the climatic conditions, which led to a large amount of noise in the spectral data. This indicated that the algorithms for extracting high-frequency information were inappropriate for the analysis of spectral information from vegetation in the Alaer region. Compared to the traditional mathematical transformations, the estimation of the relative chlorophyll content of the leaves of Korla fragrant pears based on the low-frequency information of discrete wavelets was more stable. The processing of spectral data from crops in Xinjiang is relatively complex due to the dual influence of climate and geographic location, and further research is needed to obtain rapid methods for estimating the relative chlorophyll content in the leaves of Korla fragrant pears. Although the DWT algorithm was used in this study to process the spectral data, the overall accuracy failed to reach the expected level. Therefore, there is an urgent need for more advanced and efficient algorithms to improve estimation accuracy. Meanwhile, a substantial number of scholars have already combined ground-based spectral data or vegetation physiological data with unmanned aerial vehicle (UAV) technology to predict chlorophyll content in vegetation on a large scale [38,39,40,41]. Building on existing research, it is necessary for us to further explore the application methods of remote sensing imagery data in accurately detecting vegetation information in the Alaer region. This endeavor aims to enable the measurement of chlorophyll content in large areas of vegetation with high speed, high precision, and within a short time frame.

## 5. Conclusions

The leaves of Korla fragrant pear trees were selected as the research object for this study, with the leaf spectra and the corresponding SPAD values serving as the data. The leaf spectral data were analyzed using the DWT, and the processed spectral data were correlated with the measured SPAD values to extract the characteristic bands. Finally, partial least squares was used to construct estimation models of relative chlorophyll content in the leaves. The unique geographical location and climatic conditions of the Alaer region resulted in a large amount of noise in the spectral data. The high-frequency information extracted by mathematical and discrete wavelet transforms had a limited ability to estimate the relative chlorophyll content of Korla fragrant pear leaves. Compared to the high-frequency information, the ability of low-frequency information to estimate the relative chlorophyll content was significantly greater than that of high-frequency information. The low-frequency information separated by the logarithmic transformation and the DWT significantly improved the ability to estimate the relative chlorophyll content of the leaves. The model constructed using the L1 scale of the DWT was the optimal model, with an *R*^2^ value of 0.742 and an RMSE of 0.936.

## Figures and Tables

**Figure 1 sensors-25-03552-f001:**
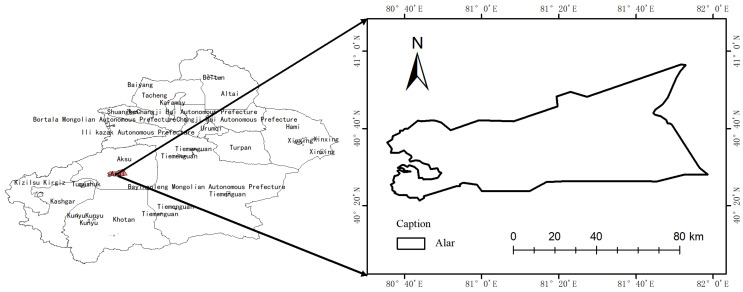
Geospatial location map of the pilot study area.

**Figure 2 sensors-25-03552-f002:**
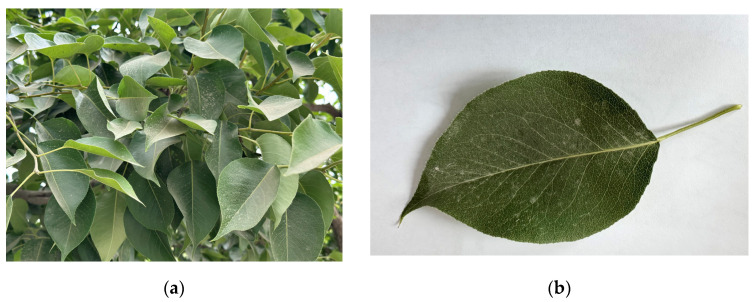
Pear tree leaf sample images. (**a**) Image of leaves from the outer canopy of a pear tree; (**b**) image of a single pear tree leaf sample.

**Figure 3 sensors-25-03552-f003:**
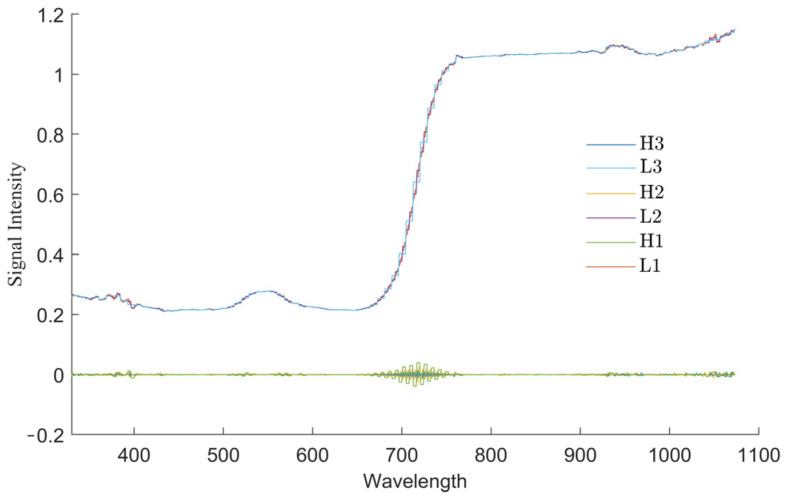
Results of discrete wavelet decomposition of raw spectral data.

**Figure 4 sensors-25-03552-f004:**
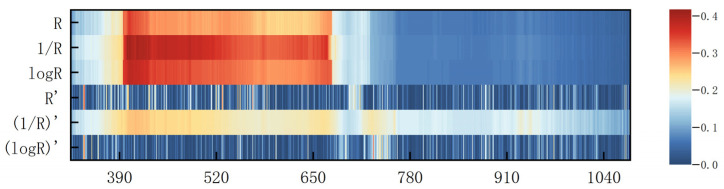
Distribution of correlation coefficients between traditional spectral transformations and the SPAD value of pear tree leaves.

**Figure 5 sensors-25-03552-f005:**
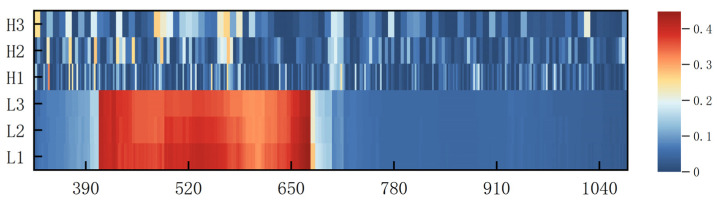
Distribution of correlation coefficients between discrete wavelet transform and SPAD value in pear tree leaves.

**Figure 6 sensors-25-03552-f006:**
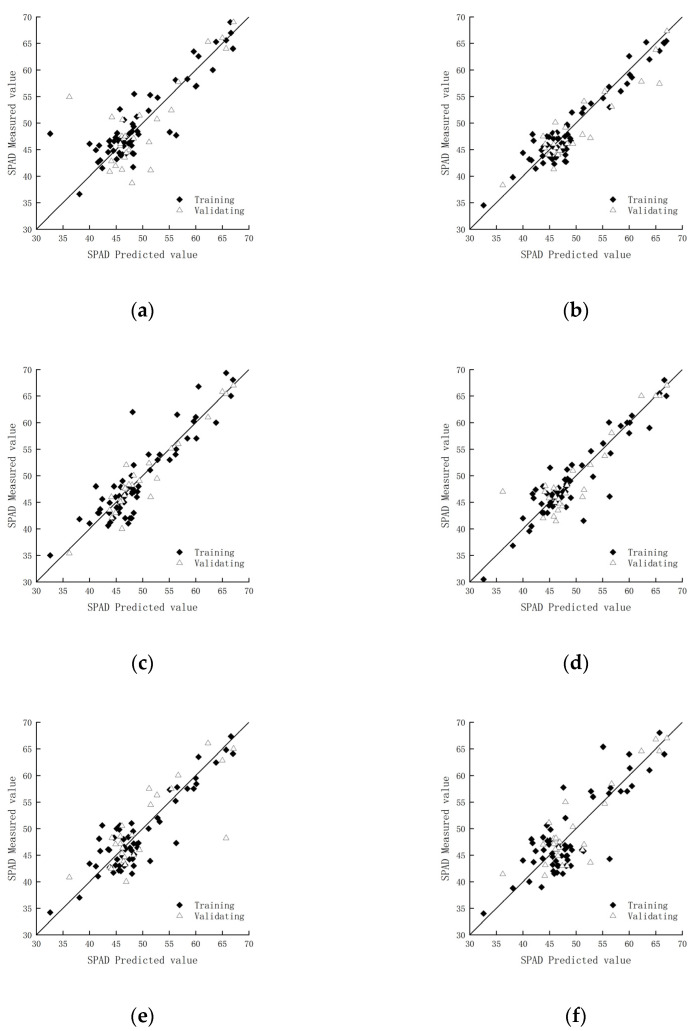
Scatter plots of measured and predicted values based on traditional mathematical transforms and the discrete wavelet L1-scale transform: (**a**) 1/R; (**b**) DWTL1; (**c**) (1/R)′; (**d**) log(R); (**e**) (log R)′; (**f**) R′.

**Table 1 sensors-25-03552-t001:** Statistical description of relative chlorophyll content in pear canopy leaf samples.

Statistical Parameter	Number of Samples	Minimum Value	Maximum Value	Average Value	Standard Deviation	RMSE	Coefficient of Variation (CV)
SPAD value	87	32.6	67.1	45.8	3.64	46.9	7.95%

**Table 2 sensors-25-03552-t002:** SPAD value estimation models for pear tree leaves based on traditional mathematical transformation.

Transformation Form	Sensitive Band (nm)	Model	Training		Validating	
			*R* ^2^	RMSE	*R* ^2^	RMSE
Original spectrum/R	455, 632	y = −0.777 + 0.002x_455_ + 0.003x_632_	0.779	1.406	0.363	0.962
Inverse transformation 1/R	501, 714	y = −0.108 + 0.013x_501_ + 0.038x_714_	0.651	1.244	0.697	1.357
Logarithmic transformation logR	472, 721	y = −1.796 + 0.037x_472_ + 0.021x_721_	0.696	1.095	0.861	0.738
First-order differential transform R′	847, 976	y = 0.004 − 0.042x_847_ + 0.003x_976_	0.834	1.821	0.785	1.172
First-order differential transform of the inverse (1/R)′	834, 998	y = −0.011 − 0.069x_834_ + 0.011x_998_	0.741	1.165	0.658	1.279
First-order differential transform of the logarithm (logR)′	825, 1034	y = 0.002 − 0.062x_825_ − 0.014x_1034_	0.063	1.986	0.885	1.925

**Table 3 sensors-25-03552-t003:** SPAD value estimation models for pear tree leaves based on the discrete wavelet transform.

Transformation Form	Sensitive Band (nm)	Model	Training		Validating	
			*R* ^2^	RMSE	*R* ^2^	RMSE
H1	474, 791	y = −0.001 − 0.032x_474_ + 0.018x_791_	0.415	1.584	0.478	1.661
H2	476, 814	y = 0.013 − 0.052x_476_ + 0.059x_814_	0.527	1.942	0.473	2.205
H3	485, 873	y = 0.005 − 0.027x_485_ + 0.203x_873_	0.337	1.672	0.235	1.389
L1	516, 724	y = 0.057 − 0.851x_516_ + 0.073x_724_	0.742	0.936	0.647	1.247
L2	534, 726	y = 0.551 − 0.861x_534_ + 0.105x_726_	0.269	1.468	0.439	1.683
L3	498, 714	y = −0.544 + 0.867x_498_ + 0.107x_714_	0.491	1.572	0.573	1.891

## Data Availability

The data involved in the study can be obtained by contacting the authors.

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
