# Peer review of "Estimation of the Relative Chlorophyll Content of Pear Leaves Based on Field Spectrometry in Alaer, Xinjiang"

_sensors, 2025, doi:10.3390/s25113552_

Round 1
Reviewer 1 Report
Comments and Suggestions for Authors
For the review comments, please refer to the PDF document.

Reviewer 2 Report
Comments and Suggestions for Authors
The manuscript titled “Research of the Inversion of Chlorophyll Content in Pear Tree Leaves Based on Hyperspectral Data” describes an approach and specific estimation model for assessing chlorophyll content in a particular plant located in a specific region. The paper includes comprehensive data on materials and methods and presents estimation results. However, some key concepts related to study motivation and interpretation of estimation results are unclearly presented. The following revisions are recommended:
Mandatory Revisions:
-
One of the key concepts involves the correlation between spectral measurement data and chlorophyll content. The proposed estimation algorithm showed good results for leaves grown in Aral City but failed in Xinjiang. This difference is attributed to noise originating from climatic conditions. Please provide more details about the nature of this noise and its dependence on climatic conditions.
-
The interpolation of results from individual leaves to whole tree crowns and larger areas at the regional scale (lines 274-279) should be addressed more critically. The mixing of spectral signals obtained within the data acquisition area in point measurements and the smallest resolved elements in spectral imaging from aerial or satellite remote sensing may complicate the analysis.
-
Please provide the rationale for selecting SPAD as the reference for chlorophyll content.
-
Selecting the optimal wavelength range and spectral band combination is a crucial and urgent task in optical spectral plant monitoring. Many researchers work on this task while seeking optimal device design or effective data processing algorithms. Indeed, optimal band combinations vary among different agricultural crops. Please address this aspect more thoroughly in the Introduction section.
Suggested Improvements:
-
Figures 3 and 4 lack axis labels.
-
Line 188: Figure 4 is incorrectly designated as Figure 1.
-
Line 277: Please correct the word “particularly.”
-
Abstract: R² is inconsistently designated as R2 – please use a uniform designation.
Round 2
Reviewer 2 Report
Comments and Suggestions for Authors
Thank you for addressing the comments. I believe that the manuscript is more balanced after revision, and the interconnections between methods, results, and discussion are clearer and more coherent.